

# Osteohistology of Late Triassic prozostrodontian cynodonts from Brazil

Jennifer Botha-Brink[1,2], Marina Bento Soares[3] and Agustín G. Martinelli[3]

[1] Department of Karoo Palaeontology, National Museum, Bloemfontein, South Africa
[2] Department of Zoology and Entomology, University of the Free State, Bloemfontein, South Africa
[3] Departamento de Paleontologia e Estratigrafia, Instituto de Geociências, Universidade Federal do Rio Grande do Sul, Porto Alegre, Brazil

Corresponding author
Jennifer Botha-Brink,
jbotha@nasmus.co.za

## ABSTRACT

The Prozostrodontia includes a group of Late Triassic-Early Cretaceous eucynodonts plus the clade Mammaliaformes, in which Mammalia is nested. Analysing their growth patterns is thus important for understanding the evolution of mammalian life histories. Obtaining material for osteohistological analysis is difficult due to the rare and delicate nature of most of the prozostrodontian taxa, much of which comprises mostly of crania or sometimes even only teeth. Here we present a rare opportunity to observe the osteohistology of several postcranial elements of the basal prozostrodontid *Prozostrodon brasiliensis*, the tritheledontid *Irajatherium hernandezi*, and the brasilodontids *Brasilodon quadrangularis* and *Brasilitherium riograndensis* from the Late Triassic of Brazil (Santa Maria Supersequence). *Prozostrodon* and *Irajatherium* reveal similar growth patterns of rapid early growth with annual interruptions later in ontogeny. These interruptions are associated with wide zones of slow growing bone tissue. *Brasilodon* and *Brasilitherium* exhibit a mixture of woven-fibered bone tissue and slower growing parallel-fibered and lamellar bone. The slower growing bone tissues are present even during early ontogeny. The relatively slower growth in *Brasilodon* and *Brasilitherium* may be related to their small body size compared to *Prozostrodon* and *Irajatherium.* These brasilodontids also exhibit osteohistological similarities with the Late Triassic/Early Jurassic mammaliaform *Morganucodon* and the Late Cretaceous multituberculate mammals *Kryptobaatar* and *Nemegtbaatar.* This may be due to similar small body sizes, but may also reflect their close phylogenetic affinities as *Brasilodon* and *Brasilitherium* are the closest relatives to Mammaliaformes. However, when compared with similar-sized extant placental mammals, they may have grown more slowly to adult size as their osteohistology shows it took more than one year for growth to attenuate. Thus, although they exhibit rapid juvenile growth, the small derived, brasilodontid prozostrodontians still exhibit an extended growth period compared to similar-sized extant mammals.

## INTRODUCTION

The non-mammaliaform cynodonts are the most derived and mammal-like clade of non-mammalian synapsids (Therapsida). They first appeared during the late Permian

(*Botha, Abdala & Smith, 2007*; *Kammerer, 2016*) and were relatively rare components of Permian ecosystems until after the Permo-Triassic mass extinction (*Hopson & Kitching, 2001*; *Abdala & Ribeiro, 2010*), when they radiated rapidly into a diverse clade with an increasingly mammalian morphology (*Luo, 2007*). The Triassic non-mammaliaform eucynodonts diverged into two major lineages, the Cynognathia and Probainognathia (*Hopson & Kitching, 2001*). Within the Cynognathia, a group of large-bodied herbivorous/omnivorous taxa known as the Gomphodontia arose during the Middle Triassic. This clade includes the families Diademodontidae, Trirachodontidae and Traversodontidae (*Crompton, 1955*; *Hopson & Kitching, 2001*; *Abdala, 2007*). The latter group includes highly diverse, globally distributed non-mammaliaform cynodonts that became especially abundant during the Late Triassic, with an apomorphic masticatory system for cutting and grinding food (*Abdala & Ribeiro, 2010*). The Probainognathia appeared during the early Late Triassic and gave rise to the Prozostrodontia, a clade that became progressively smaller-bodied during its evolution before evolving into crown group mammals during the Early Jurassic (*Liu & Olsen, 2010*; *Ruta et al., 2013*; *Martinelli & Soares, 2016*). The non-mammaliaform prozostrodontians illustrate the prior steps to the rise of Mammaliaformes and are thus important for understanding the origin and evolution of mammals. The Prozostrodontia contain several unranked taxa (e.g., *Prozostrodon*, *Therioherpeton*, *Alemoatherium*, *Botucaraitherium*), as well as two well-known major clades, the Tritylodontidae and Tritheledontidae, both closely related to Mammaliaformes (*Hopson & Kitching, 2001*; *Liu & Olsen, 2010*; *Soares, Martinelli & Oliveira, 2014*; *Martinelli et al., 2017a*; *Martinelli et al., 2017b*). The Tritylodontidae are a group of medium-sized (basal skull length, BSL ∼50–250 mm) highly specialized herbivorous prozostrodontians that arose during the Late Triassic and went extinct during the Early Cretaceous (e.g., *Clark & Hopson, 1985*; *Sues, 1986*; *Matsuoka, Kusuhashi & Corfe, 2016*). The Tritheledontidae are a group of small (BSL ∼30–70 mm), faunivorous/frugivorous forms that existed from the Late Triassic to Early Jurassic (*Kemp, 2005*; *Gow, 1980*; *Martinelli et al., 2005*; *Martinelli & Rougier, 2007*; *Soares, Schultz & Horn, 2011*).

There has been much debate regarding which group is more closely related to mammals as both groups have derived cranial and postcranial features typical of Mammaliaformes that reveals different topologies according to different phylogenetic hypotheses (e.g., *Luo, 1994*; *Hopson & Kitching, 2001*; *Kemp, 2005*; *Abdala, 2007*). However, more recently discovered taxa from Brazil (i.e., *Brasilodon quadrangularis*, *Brasilitherium riograndensis*, *Minicynodon maieri*, *Botucaraitherium belarminoi*; *Bonaparte et al., 2003*; *Bonaparte et al., 2010*; *Bonaparte, Martinelli & Schultz, 2005*; *Bonaparte, Soares & Martinelli, 2012*; *Soares, Martinelli & Oliveira, 2014*) show a closer relationship to Mammaliaformes than the aforementioned groups (*Bonaparte, Martinelli & Schultz, 2005*; *Soares, Martinelli & Oliveira, 2014*; *Martinelli et al., 2017a*; *Martinelli et al., 2017b*). Some of these species were grouped into Brasilodontidae (*Bonaparte, Martinelli & Schultz, 2005*), but their monophyly and composition (e.g., *Bonaparte, 2013*) should be tested more comprehensively, as well as the taxonomic validity of some of the members (see *Liu & Olsen, 2010*; *Martinelli & Bonaparte, 2011*; *Martinelli, 2017*). These tiny (BSL approximately ∼20–55 mm) faunivorous/frugivorous animals lived during the Late Triassic and have a set of derived

features (e.g., "triconodont"—like dentition, lack of a postorbital bar, slender horizontal ramus of the dentary and zygomatic arch, petrosal with a promontorium) that is not seen in any other non-mammaliaform cynodont, making them important taxa for understanding the early evolution of Mammaliaformes (*Kemp, 2005*; *Bonaparte, Martinelli & Schultz, 2005*; *Bonaparte, Soares & Martinelli, 2012*; *Bonaparte, 2013*; *Rodrigues, Ruf & Schultz, 2013*; *Rodrigues, Ruf & Schultz, 2014*).

The South American non-mammaliaform prozostrodontians and closely related forms (e.g., *Protheriodon*, *Candelariodon*) are particularly noteworthy as several important finds have shed light on the morphological changes that occurred prior to and during the evolution of Mammaliaformes (*Bonaparte & Barberena, 2001*; *Bonaparte, Martinelli & Schultz, 2005*; *Soares, Schultz & Horn, 2011*; *Rodrigues, Ruf & Schultz, 2013*; *Martinelli, Soares & Schwanke, 2016*; *Martinelli et al., 2017a*; *Martinelli et al., 2017b*). This remarkable record includes about 20 species of probainognathians (ecteniniids, chiniquodontids, probainognathids and prozostrodontians) from Brazil and Argentina in a fairly continuous span of ~8 million years (*Abdala & Ribeiro, 2010*; *Martinelli & Soares, 2016*). Among prozostrodontians, many taxa are tiny with some having basal skull lengths of only 20–40 mm (e.g., *Minicynodon*, *Brasilodon*, *Brasilitherium*) (*Bonaparte et al., 2010*; *Martinelli, Soares & Schwanke, 2016*). Given that they are so small and delicate, the best represented elements are jaws and teeth, with only a few partial and/or complete skulls and very few associated postcranial bones.

Apart from numerous studies on the phylogenetic relationships of non-mammaliaform cynodonts (as mentioned above), various studies have provided insight into the locomotory, masticatory, reproductive, ventilatory and physiological capacities of this clade. Differentiation of the vertebral column and an increasingly parasagittal gait during non-mammaliaform cynodont evolution resulted in improved agility and increased foraging capacity (*Kemp, 2005*; *Hopson, 2012*) (Fig. 1(1)). The presence of a bony secondary palate (Fig. 1(2)), which although not selected for endothermy (as it was more likely selected for bite strength as in dicynodont and therocephalian therapsids; *Thomason & Russell, 1986*), would have allowed them to eat and breathe simultaneously allowing more continuous ventilation to take place (*Bennett & Ruben, 1986*). Tooth differentiation and occlusion as well as reorganization of the jaw musculature facilitated radiations into new ecological niches (e.g., bucco-lingually expanded postcanines in the Gomphodontia (Fig. 1(3)); *Crompton, 1955*; *Abdala & Ribeiro, 2010*) allowing for increased energy assimilation (*Kemp, 2005*). The recovery of a juvenile (BSL 40% that of adults) with two adult specimens of the Middle Triassic trirachodontid *Trirachodon* within a burrow (NMQR 3281; *Groenewald, Welman & MacEachern, 2001*) strongly suggests that some form of extended parental care had begun to evolve within this lineage (Fig. 1(4)). The loss of the pineal foramen within the Probainognathia (Fig. 1(5)) also suggests improved reproductive timing, as well as improved thermoregulatory control (e.g., *Benoit et al., 2016*). Modifications to the neurosensory system occurred during the Triassic as well. For example, changes to the maxillary nerves in non-mammaliaform probainognathians suggests that maxillary vibrissae (i.e., whiskers) were present in some tritylodontids, the tritheledontid *Pachygenelus* and brasilodontid *Brasilitherium* indicating that hair

(perhaps not a pelage), a key mammalian feature (Fig. 1(6)), had evolved by the early Late Triassic (*Benoit, Manger & Rubidge, 2016*). An efficient counter current exchange ventilation system for the conservation of heat and water evolved within the more derived non-mammaliaform prozostrodontians as ossified maxilloturbinate bones have been found in the nasal cavity of the derived Late Triassic-Early Jurassic tritheledontid *Elliotherium* (*Crompton et al., 2017*). This system facilitates high ventilation rates in extant mammals and is a prerequisite for homeothermic endothermy (*Crompton et al., 2017*) (Fig. 1(7?)). However, ossified maxilloturbinate bones have also been found in the Late Triassic-Early Jurassic mammaliaform *Morganucodon*. Thus, this feature either evolved independently in the two lineages during the Rhaetian or it did so prior to the divergence of Tritheledontidae and Mammaliaformes during the Carnian, placing the emergence of this system earlier by some 13 million years (Fig. 1(7?)). The above examples show that there was a progressive increase in the acquisition of anatomical and physiological mammalian features during the Middle–Late Triassic. However, these studies are based on morphology and taphonomy and cannot provide insight into the growth patterns within the non-mammaliaform cynodont lineage.

Life history data, such as growth patterns and growth rates and how these change through ontogeny, can be obtained from extinct vertebrates by studying their bone microstructure or osteohistology. Given the importance of non-mammaliaform cynodonts in understanding the origin and evolution of mammalian growth patterns, numerous osteohistological studies have been conducted, but these have mostly focused on basal taxa and the more derived Cynognathia (e.g., *De Ricqlès, 1969*; *Botha & Chinsamy, 2000*; *Botha & Chinsamy, 2004*; *Botha & Chinsamy, 2005*; *Ray, Botha & Chinsamy, 2004*; *Chinsamy & Abdala, 2008*; *Botha-Brink, Abdala & Chinsamy, 2012*; *Veiga, Botha-Brink & Soares, 2018*; *Butler, Abdala & Botha-Brink, 2018*). These studies indicate that fibrolamellar bone (indicative of rapid bone deposition rates) is the dominant bone tissue type, at least during early to middle ontogeny. This bone tissue type is plesiomorphic for non-mammaliaform cynodonts, having been found in even the most basal members of the clade such as the small late Permian *Procynosuchus* (*Ray, Botha & Chinsamy, 2004*). Early Triassic basal epicynodonts remained small, but grew more rapidly than *Procynosuchus* (*Botha & Chinsamy, 2004*; *Butler, Abdala & Botha-Brink, 2018*), possibly in response to shortened life expectancies in the post-extinction environment following the Permo-Triassic mass extinction (*Botha-Brink et al., 2016*). The Middle Triassic Cynognathia generally show an increase in body size and rapid growth rates during early and mid-ontogeny, although they took longer to reach skeletal maturity compared to the Early Triassic taxa, as shown by the multiple Lines of Arrested Growth (LAG) prior to growth asymptoty (*Botha-Brink, Abdala & Chinsamy, 2012*). LAGs (represented by a cement line) and annuli (comprising narrow regions of slowly forming parallel-fibered or lamellar bone) indicate temporary cessations or decreases in bone growth respectively, and based on experiments on extant vertebrates, are known to be deposited annually during the cold or dry season (e.g., *Hutton, 1986*). The transition to multi-year growth to reproductive and skeletal maturity in the Cynognathia may be due to larger body sizes or a relaxation on the selection for early developmental times when life expectancies were higher during the Middle Triassic (*Botha-Brink et al., 2016*).

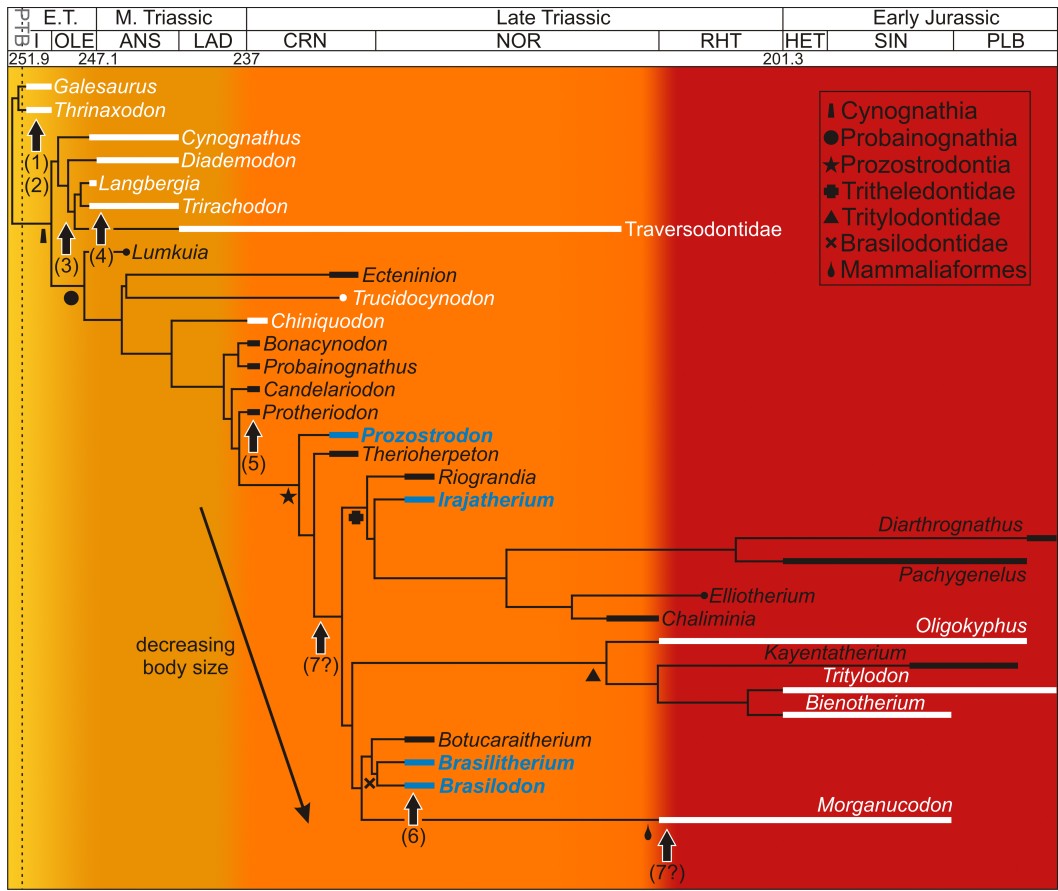

**Figure 1  Acquisition of mammalian features mapped onto a Triassic–Jurassic cynodont phylogenetic tree.** Growth patterns are known for those taxa shown in white and reveal high growth rates throughout the cynodont lineage. Taxa examined in this study appear in blue. Colors indicate increasing acquisition of mammalian characteristics during cynodont evolution (from yellow to red). Arrows indicate the possible timing of the evolution of mammalian characteristics for which morphological or behavioral data are known. (1) differentiation of vertebral column resulting in improved agility (*Kemp, 2005*), (2) bony secondary palate, allowing for more continuous ventilation (*Bennett & Ruben, 1986*), (3) increased tooth differentiation and improved tooth occlusion, allowing radiation into new ecological niches, here indicated by the Gomphodontia (*Abdala & Ribeiro, 2010*), (4) extended parental care, suggesting increased parental investment in young (*Groenewald, Welman & MacEachern, 2001*), (5) loss of pineal foramen indicating increasing thermoregulatory and reproductive control (*Benoit et al., 2016a*), (6) possible evolution of maxillary vibrissae indicating improved sensory structures (*Benoit, Manger & Rubidge, 2016b*), (7) ossified maxillary turbinates indicating efficient counter current exchange system for the conservation of heat and water (*Crompton et al., 2017*). Phylogeny taken from *Ruta et al. (2013)* and *Martinelli et al. (2017a)*; *Martinelli et al. (2017b)*. Dates taken from *Gradstein, Ogg & Hilgen (2012)* and *Martinelli et al. (2017a)*; *Martinelli et al. (2017b)*. Abbreviations: ANS, Anisian; CRN, Carnian; E.T., Early Triassic; HET, Hettangian; I, Induan; LAD, Ladinian; M., Middle; OLE, Olenekian; PLB, Pliensbachian; RHT, PTB, Permo-Triassic Boundary; Rhaetian; SIN, Sinemurian.

Exceptions to this are the relatively smaller-bodied traversodontids *Andescynodon* and *Massetognathus* that grew relatively more slowly (see *Chinsamy & Abdala, 2008* for a detailed review). All non-mammaliaform cynodonts exhibit slower forming bone tissues, such as parallel-fibered or lamellar bone, upon departure from the juvenile stage

(*Botha-Brink, Abdala & Chinsamy, 2012*), with growth asymptoty being represented by an External Fundamental System (*Cormack, 1987*); or outer circumferential lamellae by other authors, e.g., *Chinsamy-Turan (2005)*. An EFS is a peripheral region of avascular or very poorly vascularized slowly forming bone tissue that often contains multiple, closely spaced LAGs, and indicates that skeletal growth has essentially ceased. Although it is rare to find senescent individuals in the fossil record, the presence of this feature in at least some taxa indicates that, similar to other vertebrates (*Woodward, Horner & Farlow, 2011*; *Lee et al., 2013*), non-mammaliaform cynodonts underwent determinate growth.

Although much research has been conducted on the osteohistology of basal non-mammaliaform cynodonts and the more derived Cynognathia, relatively little attention has been given to the Probainognathia (*De Ricqlès, 1969*; *Ray, Botha & Chinsamy, 2004*; *Botha-Brink, Abdala & Chinsamy, 2012*). Apart from the tritylodontid *Tritylodon* (*Botha-Brink, Abdala & Chinsamy, 2012*), only brief descriptions of poorly preserved material have been provided for *Chiniquodon*, *Bienotherium* (*De Ricqlès, 1969*) and *Trucidocynodon* (*Botha-Brink, Abdala & Chinsamy, 2012*). These taxa represent either the early (*Trucidocynodon*, *Chiniquodon* from the Carnian) or later part of non-mammaliaform probainognathian evolution (*Bienotherium* from the Rhaetian-Hettangian, *Oligokyphus* from the Rhaetian-Pliensbachian) (Fig. 1). We describe here the limb bone osteohistology of four non-mammaliaform prozostrodontian probainognathian taxa from the Carnian and Norian of Brazil. We assess their life histories using limb bone osteohistology and compare their growth patterns with those of other non-mammaliaform cynodonts (e.g., basal cynodonts, cynognathians, basal probainognathians) and Mammaliaformes, thus filling a knowledge gap in the cynodont-mammal transition.

## MATERIAL AND METHODS

### Material

Our sample comprises *Prozostrodon brasiliensis* (estimated BSL ∼75–80 mm), which is one of the most basal prozostrodontians (*Bonaparte & Barberena, 2001*; *Liu & Olsen, 2010*; *Martinelli, Soares & Schwanke, 2016*), the tritheledontid *Irajatherium hernandezi* (*Martinelli et al., 2005*; *Oliveira, Martinelli & Soares, 2011*) (estimated maximum BSL 80 mm), and the brasilodontids *Brasilodon quadrangularis* and *Brasilitherium riograndensis* (both with a maximum estimated BSL of 40 mm). The BSLs of *Brasilodon* and *Brasilitherium* are taken from specimens that were assigned to each of these taxa by *Bonaparte (2013)*, *Bonaparte, Martinelli & Schultz (2005)* and *Bonaparte, Soares & Martinelli (2012)*. However, *Liu & Olsen (2010)* and *Martinelli & Bonaparte (2011)* suggested that these two taxa should be synonymised and that the specimens of both taxa represent different ontogenetic stages of one taxon (see also *Martinelli, 2017*). With the exception of one specimen (UFRGS-PV 1043; *Bonaparte et al., 2010*; *Bonaparte, Soares & Martinelli, 2012* used also in other contributions), most specimens referred to *Brasilitherium* are small-sized compared to the *Brasilodon* ones. Some authors have maintained the use of both taxa in their analyses (e.g., *Bonaparte, 2013*; *Soares, Martinelli & Oliveira, 2014*; *Martinelli et al., 2017a*; *Martinelli et al., 2017b*), but studies are currently being undertaken (AGM, MBS)

that will shed light on this issue. A consensus has yet to be reached and thus, the taxa are treated separately here, with the caveat that the osteohistology of these two taxa may be shown to represent one species in the future.

The material of *Prozostrodon brasiliensis* includes a proximal portion of a left humerus and complete left femur of the holotype UFRGS-PV-248-T (the only known specimen with postcranial material). UFRGS-PV-248-T was found 200 m northwest of the Cerriquito Hill, in a road cut on highway BR-287, in the municipality of Santa Maria, Rio Grande do Sul State, Brazil (*Bonaparte & Barberena, 2001*). It comes from the *Hyperodapedon* Assemblage Zone (AZ) of the Candelária Sequence (*Horn et al., 2014*), Santa Maria Supersequence (*Zerfass et al., 2003*), which is considered to be Late Carnian in age based on biostratigraphical correlations with the Ischigualasto Formation of Argentina (e.g., *Langer, 2005*) and radiometric dating using U-Pb zircon geochronology (233.23 ± 0.73 million years ago (*Langer, Ramezani & Da Rosa, 2018*).

The material of *Irajatherium hernandezi* comprises an almost complete humerus (with only the proximal end missing) (UFRGS-PV-1072-T), which was described by *Oliveira, Martinelli & Soares (2011)*. The material of *Brasilodon quadrangularis* consists of the shaft of an ulna (the articular proximal portion was preserved separately) associated with a skull and jaws (UFRGS-PV-765-T). It was figured in *Bonaparte, Martinelli & Schultz (2005: Fig. 6)*. The *Brasilitherium riograndensis* material consists of two specimens: UFRGS-PV-1308-T (small-sized specimen), preserved in a sandstone block with a radius, ulna, femur, tibia, and fibula (associated with a lower jaw and maxilla referred to *Brasilitherium*, which has not been formally published); UFRGS-PV-1043-T (large-sized specimen), a proximal portion of the left femur, which is associated with a skull, jaws and other postcranial elements (e.g., *Bonaparte et al., 2010*; *Bonaparte, Soares & Martinelli, 2012*; *Rodrigues, Ruf & Schultz, 2013*; *Rodrigues, Ruf & Schultz, 2014*; *Ruf et al., 2014*).

All the specimens of *Irajatherium*, *Brasilitherium* and *Brasilodon* come from the outcrop known as Linha São Luiz Site, located approximately 3 km north of Faxinal do Soturno city, Rio Grande do Sul State, Brazil. They come from the *Riograndia* AZ (above the *Hyperodapedon* AZ) of the Candelária Sequence (*Horn et al., 2014*), Santa Maria Supersequence (*Zerfass et al., 2003*), of Norian age based on a radiometric U-Pb date of the Linha São Luiz Site (maximum of 225.42 ± 0.25 million years ago; *Langer, Ramezani & Da Rosa, 2018*) and biostratigraphical correlation with the Ischigualasto and Los Colorados formations of Argentina (e.g., *Martínez et al., 2013*).

## Methods

All the elements were measured and photographed prior to thin sectioning. Although mostly fragmentary material was thin sectioned (given the rarity of the taxa), all the bones were associated with diagnostic cranial material. The elements were serially sectioned where possible, but some of the bones were so small that only one or two sections could be recovered from each bone. The thin sectioning process was conducted at the National Museum, Bloemfontein, South Africa. The bones were embedded in the clear Struers Epofix epoxy resin (Cleveland, OH, USA) under vacuum. Once set, the resin blocks were serially sectioned using a Struers Accutom-100 cutting and grinding machine. Each thick

section was adhered to a frosted glass slide using the Struers Epofix resin. These sections were then ground to a thickness of approximately 60 μm using the Struers Accutom-100 and polished manually using Struers Accutom-100 cutting oil. The resulting thin sections were then digitally rendered under ordinary, polarized (PL), and cross-polarized light (CPL), using a Nikon Eclipse Ci POL polarizing microscope and DS-Fi3 digital camera in NIS-Elements 4.6 (Nikon Corp., Tokyo, Japan). Osteohistology terminology follows that of *Francillon-Vieillot et al. (1990)* and *De Ricqlès et al. (1991)*.

## RESULTS

### *Prozostrodon brasiliensis*

A left humerus (UFRGS-PV248a-T) comprising the proximal midshaft (the region of the deltopectoral crest) and epiphysis were thin sectioned (Figs. 2A–2C). In transverse section, the bone contains a relatively small medullary cavity surrounded by a thick compact cortex (Fig. 2A). A few thick bony trabeculae traverse the otherwise clear medullary cavity. Several large resorption cavities in the perimedullary region are surrounded by thick layers of endosteal lamellae. A few of these resorption cavities extend into the area of the deltopectoral crest (Fig. 2A). Small secondary osteons, demarcated by their cement lines, are rare and limited to the innermost cortex in the midshaft, but are more abundant in the metaphyseal region. The compact cortex is comprised of a woven-fibered matrix with abundant, large, globular, haphazardly arranged osteocyte lacunae and numerous vascular canals (Figs. 2B and 2C) forming a fibrolamellar bone tissue complex (defined by the presence of a woven-fibered bone matrix associated with primary osteons; *Francillon-Vieillot et al., 1990*). Most of the canals form poorly developed longitudinally-oriented primary osteons in radial rows, but some radiate transversely, particularly in the region of the deltopectoral crest (Fig. 2A), and others a reticular pattern, especially at the bone periphery, further down the midshaft. The fibrolamellar bone is interrupted by two regions of slower forming parallel-fibered bone tissue (Fig. 2B). The first is a thick region (16 μm) of less vascularized parallel-fibered bone tissue midway through the cortex. The osteocyte lacunae are more uniformly distributed in this region (i.e., there is no evidence of static osteogenesis; *Stein & Prondvai, 2013*). The parallel-fibered bone matrix becomes increasingly organized within its center to form slowly forming lamellar bone with flattened osteocyte lacunae arranged parallel to one another. A LAG (Figs. 2A and 2B), which becomes an annulus in parts (Fig. 2C), indicating a temporary cessation or decrease in growth rate respectively, traverses the middle of this slower growing region. A thin zone of more vascularized fibrolamellar bone follows from this region, which is then followed by a second, thinner region (7 μm) of parallel-fibered bone in the outer cortex. The bone tissue returns to highly vascularized fibrolamellar bone at the subperiosteal surface with a reticular or radiating vascular network (Fig. 2B), showing that the bone was still actively growing at the time of death. Sharpey's fibers, indicating areas of muscle attachment, were observed in the proximal metaphysis on the dorsal side of the bone (Fig. 2C), possibly for the insertion of the latissimus dorsi or teres minor muscles (*Jenkins, 1971*).

In transverse section, the left femur (UFRGS-PV-248b-T) comprises a small medullary cavity surrounded by a thick compact cortex (Fig. 2D). The medullary cavity is completely

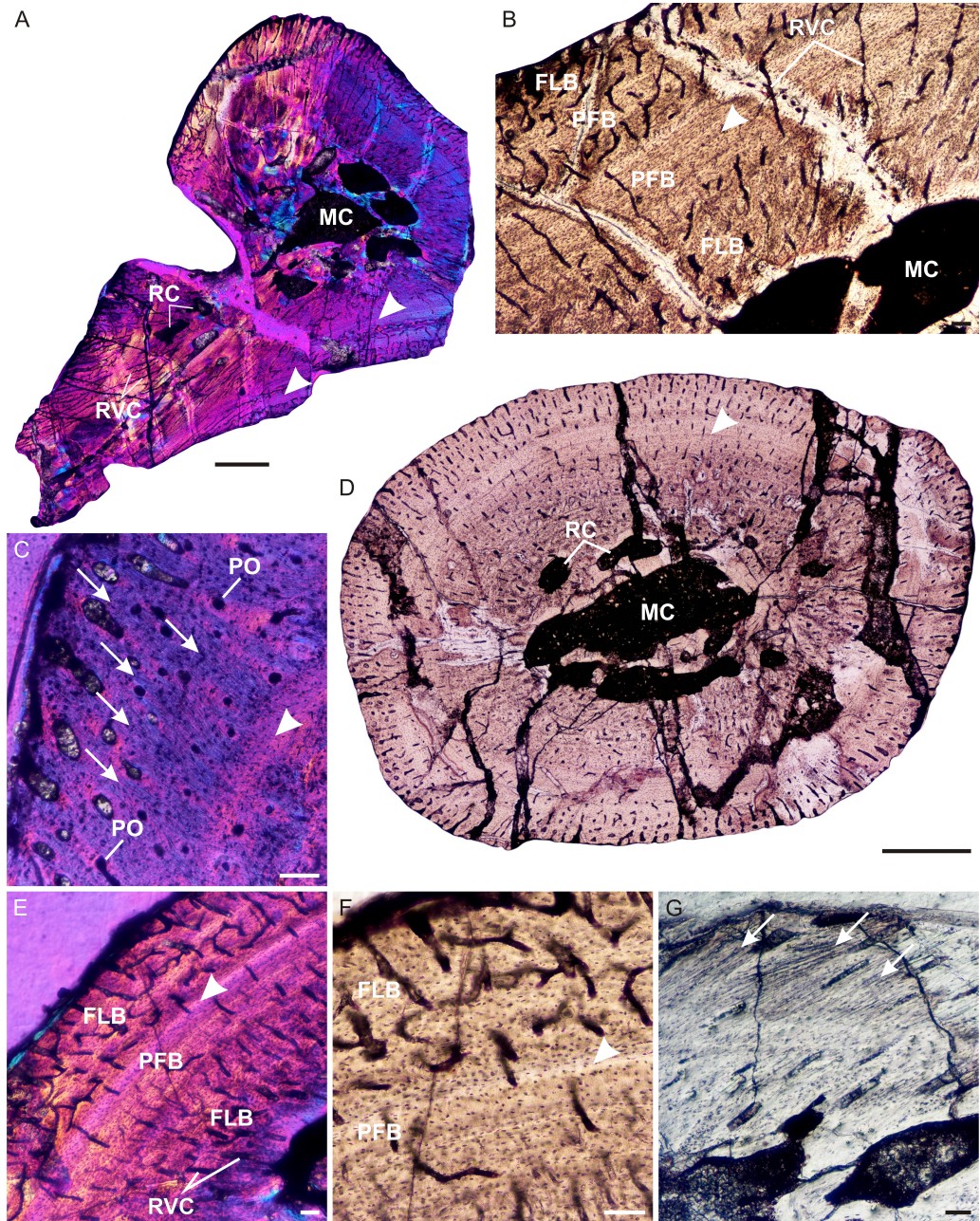

**Figure 2  Limb bone osteohistology of *Prozostrodon brasiliensis* UFRGS-PV-248-T.** (A) Whole cross-section of humerus UFRGS-PV-248a-T in CPL showing a relatively thick cortex and two regions of parallel-fibered bone (arrowheads). (B) High magnification of the humerus showing alternating fibrolamellar and parallel-fibered bone. Arrowhead indicates a LAG within the parallel-fibered bone region. (C) Sharpey's fibers (arrows) in the humerus indicating an area of muscle insertion and a LAG (arrowhead) in the mid-cortex. (D) Whole cross-section of femur UFRGS-PV-248b-T showing a relatively thick cortex and a LAG (arrowhead) running through the wide zone of parallel-fibered bone. (E) High magnification of the femur showing alternating fibrolamellar (continued on next page...)

**Figure 2 (…continued)**
and parallel-fibered bone with an annulus (arrowhead). (F) Peripheral reticular vascular canals in the fibrolamellar bone of the femur. Arrowhead indicates an annulus. (G) Longitudinal section of the femur showing Sharpey's fibers (arrows) indicating an area of muscle insertion. Abbreviations: FLB, fibrolamellar bone; MC, medullary cavity; PFB, parallel-fibered bone; PO, primary osteons; RC, resorption cavities; RVC, radiating vascular canals. Scale bars equal 100 μm, apart from A and D, which equal 1,000 μm.

clear and is only traversed by a few thick broken trabeculae further away from the midshaft. Several large resorption cavities can be found in the perimedullary region (Fig. 2D). Secondary osteons are small and rare, and limited to the innermost cortex, similar to the humerus. The compact cortex is similar to the humerus, i.e., it comprises highly vascularized fibrolamellar bone with mostly short radiating vascular canals and longitudinally-oriented primary osteons in the inner cortex and short radiating canals at the subperiosteal surface (Fig. 2D), which become more reticular further down the midshaft (Figs. 2E and 2F). The fibrolamellar bone is also interrupted by parallel-fibered bone tissue (10 μm thick) midway through the cortex, similar to the humerus. It appears as two regions in some areas (similar to the humerus), but merges to form one zone in others. A LAG (Fig. 2D), which becomes an annulus in parts (Figs. 2E and 2F), can be seen running through parts of this region. Rapid growth resumes after this slow growing region in the form of rapidly growing fibrolamellar bone, indicating active growth at the time of death. Sharpey's fibers, indicating areas of muscle insertion, were observed on the dorsomedial and ventrolateral sides of the midshaft, possibly for the origin of the femoro-tibialis and adductor muscles, respectively, as well as on the ventral side of the proximal epiphysis (Fig. 2G), possibly for the insertion of the ilio-femoralis or adductor muscles (*Jenkins, 1971*). The femoral longitudinal sections reveal a similar bone tissue pattern to the transverse sections. The vascular canals in both proximal and distal epiphyses stop just beneath the bone surfaces, which are themselves capped by a thin layer of calcified cartilage.

### *Irajatherium hernandezi*

The midshaft and distal end of a left humerus (UFRGS-PV-1072-T) were available for thin sectioning (Figs. 3A–3D). The bone tissues are not particularly well preserved and appear to have been infiltrated with some kind of mineral. However, although parts appear to be diagenetic, the main features of the bone tissue can be clearly observed from several well preserved patches of primary bone. In transverse section, the bone comprises a small medullary cavity traversed by a few thick trabeculae (Fig. 3A). Secondary remodeling is limited to a few small secondary osteons in the inner cortex. The surrounding thick, compact cortex consists of an inner cortex of fibrolamellar bone with mostly longitudinally-oriented primary osteons (in radiating rows in a few patches) and a few short radiating canals (Figs. 3B and 3C). The outer third of the cortex gives way to parallel-fibered bone and in places lamellar bone (Figs. 3A and 3B), with more uniformly distributed osteocyte lacunae. At least one LAG (Fig. 3A) is present in this thick region, but there appears to be more (Figs. 3B and 3D). Confirmation is difficult, however, as the diagenesis appears to have highlighted the individual lamellae of the bone tissue. This would then suggest the presence of lamellar bone, where the individual layers of this bone tissue type have been

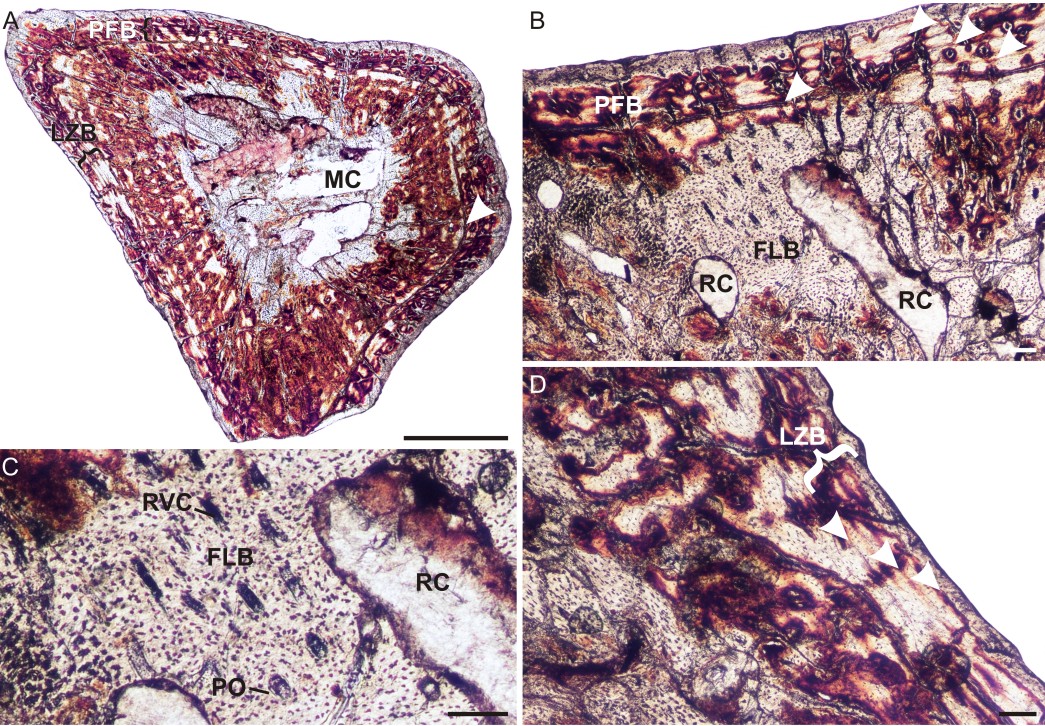

**Figure 3** **Humeral osteohistology of *Irajatherium hernandezi* UFRGS-PV-1072-T.** (A) Whole cross-section showing a relatively thick cortex. A LAG (arrowhead) marks the beginning of a slower growing zone of parallel-fibered and lamellar-zonal bone. (B) Midcortical fibrolamellar bone is followed by lamellar-zonal bone with at least one LAG, possibly more (arrowheads). (C) High magnification showing longitudinally-oriented primary osteons in a woven-fibered matrix. (D) Another region of the outer cortex showing lamellar-zonal bone (brackets) with several possible LAGs (arrowheads). Abbreviations: FLB, fibrolamellar bone; LZB, lamellar-zonal bone; MC, medullary cavity; PFB, parallel-fibered bone; PO, primary osteon; RC, resorption cavity; RVC, radial vascular canal. Scale bars equal 100 µm.

highlighted by the mineral infiltration, and indicates slow bone deposition. The osteocyte lacuna density is lower and their distribution more uniform in this region (Fig. 3D) compared to that seen in the inner and mid-cortex (Fig. 3C). The distal longitudinal sections revealed similar bone tissue patterns to the midshaft, but there are clearer views of numerous, large, globular osteocyte lacunae in the metaphysis, possibly as a result of ongoing bone activity during remodeling in this area.

Despite the uncertain number of LAGs, the features listed above do indicate a transition to slower growth. The slow growing zone may indicate an External Fundamental System (EFS), but the presence of vascular canals within the slower growing region, as well as at the subperiosteal surface suggests that the bone was still growing (although more slowly) and that maximum size had yet to be reached. Although not fully grown, however, this individual was no longer a juvenile as the overall growth rate had decreased indicating that it was a subadult that had reached the growth deceleration phase.

### *Brasilodon quadrangularis*

The ulna of *Brasilodon* (UFRGS-PV-765-T) contains a large clear medullary cavity surrounded by a thin compact cortex (Fig. 4A). A thin layer of endosteal bone with flattened, parallel osteocyte lacunae lines the medullary cavity on the medial side of the bone. The bone tissue consists primarily of woven-fibered bone with large, globular, haphazardly arranged osteocyte lacunae and a few poorly developed tiny primary osteons (Figs. 4B and 4C), with a small patch of parallel-fibered bone with more uniformly distributed osteocyte lacunae (Fig. 4D). On the medial side of the bone there is a peripheral region that contains a few flattened osteocyte lacunae and differs in color from the rest of the cortex under cross polarized light. This region of lamellar bone may represent an annulus (Fig. 4D). A few tiny secondary osteons could be observed within the olecranon process (Fig. 4A).

### *Brasilitherium riograndensis*

The radius, ulna, femur, tibia and fibula of the smaller individual of *Brasilitherium* were thin sectioned (UFRGS-PV-1308-T). The radius (UFRGS-PV-1308a-T) contains a large clear medullary cavity surrounded by poorly vascularized parallel-fibered bone (Fig. 5A). The osteocyte lacunae are globular, but arranged uniformly throughout the cortex. The vascular canals are all tiny and simple. Neither primary nor secondary osteons are present. There is no change in tissue type towards the subperiosteal surface and growth marks are absent.

The ulna (UFRGS-PV-1308b-T) has a very large clear medullary cavity and thin compact cortex. The bone tissues are simple, similar to the radius, but the flattened, parallel osteocyte lacunae indicate the presence of lamellar bone instead of parallel-fibered bone. In some patches, the individual lamellae can be clearly seen. Vascularization is simple, similar to the radius, although slightly more vascularized. One lamella, towards the subperiosteal surface, appears to be more prominent than the others, and is different in color under cross-polarized light compared to the other lamellae (Fig. 5B). It may represent a LAG, but this cannot be confirmed as it only appears in one corner of the bone.

The femur of this same individual (UFRGS-PV-1308c-T) also contains a large, free medullary cavity and thin cortex (Fig. 5C). A few broken bony trabeculae were observed in the center of the cavity. The bone tissue contains a mixture of types, ranging from woven-fibered to parallel-fibered to lamellar in various regions (Figs. 5C and 5D). Some tiny patches contain globular, haphazardly arranged osteocyte lacunae and a few tiny, poorly developed primary osteons, indicating fibrolamellar bone. Other areas contain more organized osteocyte lacunae some of which are flattened, indicating parallel-fibered and lamellar bone. In these regions, the vascular canals are simple. Although, not highly vascularized, the canals are more abundant than the radius and ulna. There is no evidence of a decrease in vascularization at the subperiosteal surface, indicating continued growth. A few tiny secondary osteons were observed in the innermost cortex. Growth marks were not observed.

The tibia (UFRGS-PV-1308d-T) is similar to the femur, with a large open medullary cavity and a mixture of bone tissue types in the compact cortex (Fig. 5E). Patches of

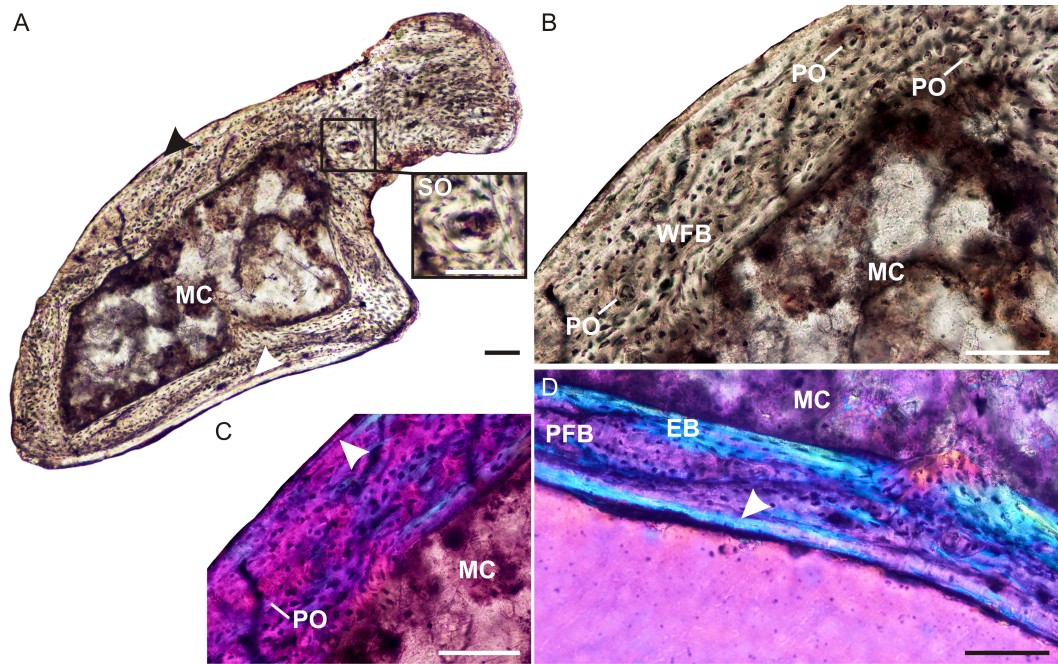

**Figure 4  Ulna osteohistology of *Brasilodon quadrangularis* UFRGS-PV-765-T.** (A) Whole cross-section showing a relatively thin cortex. Arrowheads indicate an annulus. Insert shows a high magnification of a secondary osteon. (B) High magnification of woven-fibered bone with several poorly developed primary osteons. (C) Poorly developed fibrolamellar bone in CPL, arrowhead indicates a growth mark. (D) High magnification of the annulus (arrowhead) at the bone periphery. Abbreviations: EB, endosteal bone; PFB, parallel-fibered bone; MC, medullary cavity; PO, primary osteon; SO, secondary osteon; WFB, woven-fibered bone. Scale bars equal 100 μm.

woven-fibered bone with a few primary osteons (Fig. 5F) indicating the presence of poorly developed fibrolamellar bone, is mixed with areas of parallel-fibered bone. One tiny secondary osteon was observed in the outer cortex. Growth marks are absent.

The fibula (UFRGS-PV-1308e-T) contains an open medullary cavity and a relatively thick cortex. The compact cortex comprises primarily avascular lamellar bone tissue (Fig. 5G). There is a patch of globular osteocyte lacunae towards the end of the bone in the region of the metaphysis. A few small vascular canals are also present in this region. The osteocyte lacunae comprise a mixture of both globular and flattened bodies. Growth marks were not observed.

The larger femur (UFRGS-PV-1043-T) from the second *Brasilitherium* individual also contains a large free medullary cavity and relatively thin cortex (Figs. 5H–5J). One large resorption cavity was observed towards the direction of the greater trochanter, otherwise secondary remodeling is limited to a few tiny secondary osteons in the innermost cortex. There are small patches of woven-fibered bone with haphazardly arranged osteocyte lacunae in the inner cortex, but they become increasingly uniformly distributed towards the subperiosteal surface to form parallel-fibered bone and even lamellar bone in places (Fig. 5I). Vascularization is moderate and comprises some small longitudinally-oriented primary osteons and simple canals. A LAG runs through the mid-cortex on the posterior

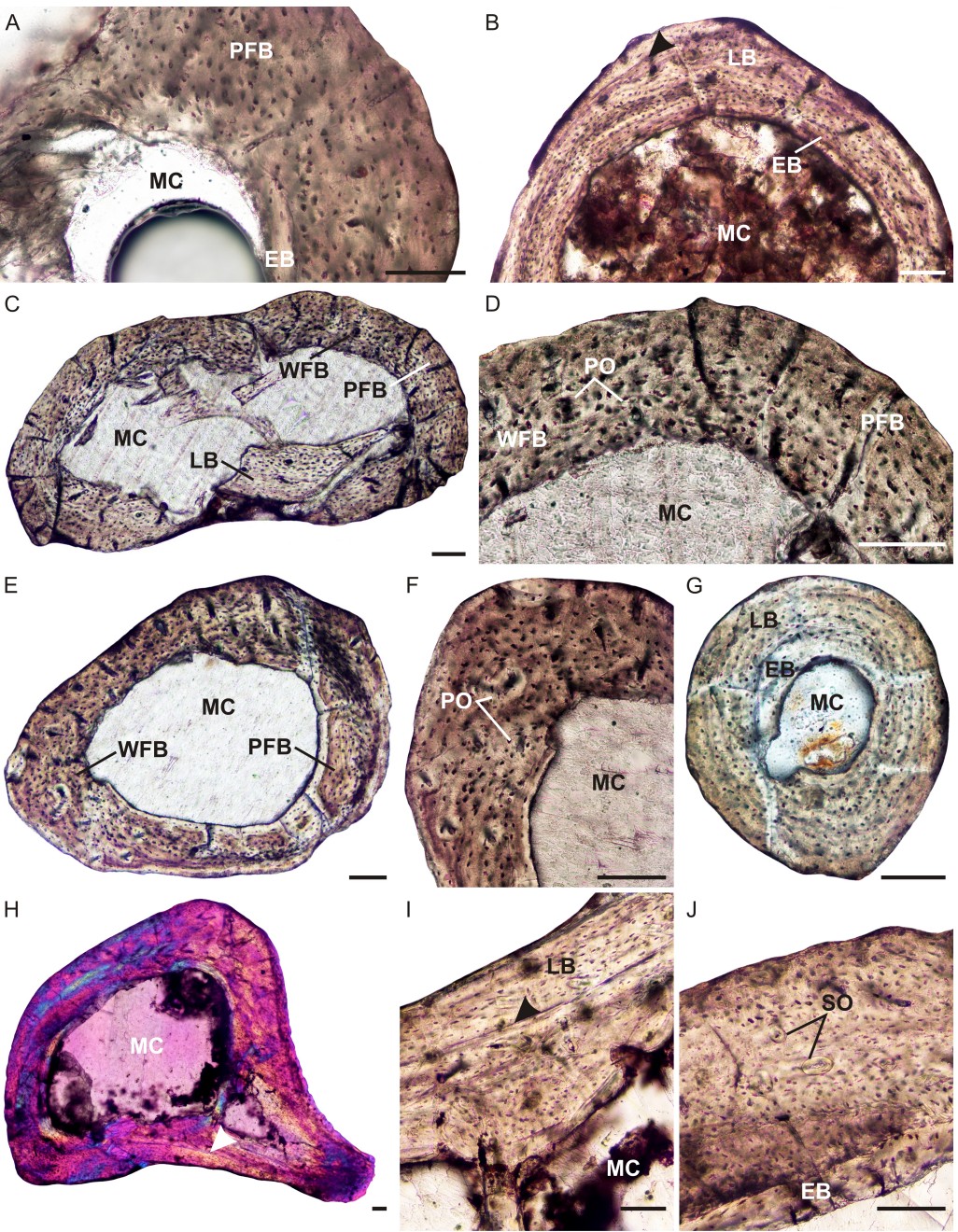

**Figure 5  Limb bone osteohistology of *Brasilitherium riograndensis*.** (A) Radius UFRGS-PV-1308a-T showing poorly vascularized parallel-fibered bone tissue. (B) Ulna UFRGS-PV-1308b-T showing lamellar bone, arrowhead indicates a possible growth mark. (C) Femur UFRGS-PV-1308c-T showing patches of woven-fibered, parallel-fibered and lamellar bone. (D) High magnification of the femur showing haphazardly arranged osteocyte lacunae (left) and more organized osteocyte lacunae (right). (E) Tibia UFRGS-PV-1308d-T showing patches of woven-fibered and parallel-fibered bone. (F) High magnification of the tibia showing several poorly developed primary osteons. (G) Fibula UFRGS-PV-1308e-T showing avascular lamellar bone. (H) Femur UFRGS-PV-1043-T in CPL showing a (continued on next page...)

**Figure 5 (…continued)**
mid-cortical LAG (arrowhead). (I) High magnification of the LAG (arrowhead) followed by slowly forming lamellar bone. (J) High magnification of femur UFRGS-PV-1043-T showing haphazardly arranged osteocyte lacunae and a few small secondary osteons. Abbreviations: EB, endosteal bone; MC, medullary cavity; PFB, parallel-fibered bone; LB, lamellar bone; PO, primary osteon; SO, secondary osteon; WFB, woven-fibered bone. Scale bars equal 100 µm.

side of the bone, although it should be noted that it cannot be traced around the whole bone. There is no evidence of an EFS, but the increased prevalence of parallel-fibered bone at the subperiosteal surface suggests an overall decrease in growth rate and a departure from the juvenile stage. Although not fully grown, this individual was likely a subadult when it died.

## DISCUSSION

*Prozostrodon brasiliensis* represents one of the most basal members of the Prozostrodontia (*Liu & Olsen, 2010*; *Martinelli et al., 2017a*; *Martinelli et al., 2017b*) and provides a good example of the growth patterns that existed during the early evolution of this clade. The presence of well vascularized, rapidly forming fibrolamellar bone is similar to that found in all other non-mammaliaform cynodonts, including one of the most basal members, *Procynosuchus*, which exhibits fibrolamellar bone during the early stages of ontogeny (*Botha-Brink, Abdala & Chinsamy, 2012*). The temporary decrease and cessation in growth shows that *Prozostrodon* grew cyclically as did most of the gomphodont cynodonts (*Botha-Brink, Abdala & Chinsamy, 2012*). This pattern differs from that observed in the basal probainognathians *Chiniquodon* (*De Ricqlès, 1969*) and *Trucidocynodon* (*Botha-Brink, Abdala & Chinsamy, 2012*), which exhibit sustained growth. However, it should be noted that the elements examined by previous authors do not represent mature individuals (*Botha-Brink, Abdala & Chinsamy, 2012*; J Botha-Brink, pers. obs., 2015), casting doubt as to whether these taxa underwent sustained growth to skeletal maturity. More material is required to adequately deduce the growth patterns of these two taxa.

The high incidence of longitudinally-oriented primary osteons in the bone tissues of *Prozostrodon*, compares well with the Early Jurassic tritylodontid *Tritylodon longaevus* (*Botha-Brink, Abdala & Chinsamy, 2012*), although the reticular patches are also similar to the Early Triassic basal epicynodont *Thrinaxodon liorhinus* (*Botha-Brink, Abdala & Chinsamy, 2012*). Vascularization is higher than that observed in *Chiniquodon*, which contains fewer longitudinally-oriented primary osteons (J Botha-Brink, pers. obs., 2015), but less vascularized than *Trucidocynodon*, which exhibits a plexiform vascular arrangement (*Botha-Brink, Abdala & Chinsamy, 2012*). The higher vascularization in *Trucidocynodon* may be related to its larger body size (BSL 188 mm, *Oliveira, Soares & Schultz, 2016*) as within a given clade larger bodied taxa tend to grow more quickly than their smaller relatives (*Case, 1978*).

It is clear from the highly vascularized, rapidly deposited bone tissues that *Prozostrodon* grew quickly during the favorable growing season. The expansive width of the slow growing parallel-fibered zone is noteworthy, however, as this differs from the gomphodonts that

have been studied to date (i.e., *Diademodon, Trirachodon, Langbergia, Andescynodon, Traversodon, Gomphodontosuchus, Protuberum, Scalenodontoides, Exaeretodon*; *Botha & Chinsamy, 2000*; *Botha & Chinsamy, 2004*; *Chinsamy & Abdala, 2008*; *Botha-Brink, Abdala & Chinsamy, 2012*; *Veiga, Botha-Brink & Soares, 2018*), all of which express LAGs or narrow annuli of parallel-fibered or lamellar bone. The broadness of this slow growing zone indicates that either the animal was able to continue growing through much of the cold or dry season, only ceasing when the LAG was deposited, or that it experienced a particularly long and harsh unfavorable growing season and thus, grew slowly for longer under these conditions.

Although UFRGS-PV-248-T is the largest known specimen of *Prozostrodon*, the presence of continued, active rapid growth at the periphery of the bones indicates that it was not a mature individual. There is no overall deceleration in growth and thus, it had not likely attained reproductive maturity before it died. A deceleration in growth rate typically accompanies the onset of reproductive maturity in extant mammals that take several years to reach a large body size (*Owens, Dubeski & Hanson, 1993*; *Köhler et al., 2012*), but may occur after asymptotic size is reached in fast growing, very small mammals (e.g., the long-clawed shrew, *Nesterenko & Ohdachi, 2001*; grey mouse lemur, *Castanet et al., 2004*) or prior to this transition in very large taxa (e.g., elephant, *Lee & Werning, 2008*). Although the maximum known BSL of *Prozostrodon*, which is 80 mm, does not represent a fully grown individual, the taxon is not likely to have reached the large body sizes of extant mammals that take several years to reach reproductive maturity, such as ruminants (two to four years), the hippopotamus (10 years) or elephant (15 years) (*Miller & Zammuto, 1983*; *Promislow & Harvey, 1990*; *Köhler et al., 2012*). *Prozostrodon* falls within the size range of small extant mammals that typically attain reproductive maturity within one year (*Miller & Zammuto, 1983*; *Promislow & Harvey, 1990*). However, given the mid-cortical growth mark in the *Prozostrodon* material and continued rapid growth, it is likely that this taxon took longer than one year to become reproductively mature.

To date, *Irajatherium hernandezi* is the only tritheledontid cynodont for which the limb bones have been studied histologically. The bone tissues revealed rapidly forming fibrolamellar bone with similar vascular arrangements to that seen in *Tritylodon* (*Botha-Brink, Abdala & Chinsamy, 2012*). Although the material is poorly preserved, peripheral interruptions in growth (at least one LAG) can be clearly seen. The presence of vascular canals at the subperiosteal surface indicates that growth continued, but the canals are fewer and smaller compared to those in the mid and inner cortex and a slower forming tissue (lamellar bone) appears in this region, indicating a decrease in growth rate. The presence of at least one LAG, however, indicates that growth was cyclical. *Botha-Brink, Abdala & Chinsamy (2012)* suggested that *Tritylodon* generally grew in a sustained manner as growth marks were rarely found in the material studied, but the presence of a double LAG in a radius and fibula belonging to specimen BP/1/5167 indicates that *Tritylodon* was capable of cyclical growth. Upon further investigation it was found that much of the *Tritylodon* material that has been thin sectioned represents juveniles and the presence of continued growth after the LAG in the bones of BP/1/5167 indicates that it was just over a year old at the time of death and not fully grown (J Botha-Brink, pers. obs., 2018). Furthermore,

a mid-cortical LAG is present in the tritylodontid *Bienotherium* femur described by *De Ricqlès (1969)*, which presents continued growth following this growth mark and then a wide region of very poorly vascularized lamellar bone tissue that may represent an EFS (J Botha-Brink, pers. obs., 2015). This suggests that tritylodontids underwent cyclical growth during mid-ontogeny.

In contrast, the *Irajatherium* material reveals that a slower growth phase had been reached and was thus not likely to have grown significantly larger than the maximum known BSL of 80 mm. This suggests that it was a smaller taxon than *Tritylodon* (maximum known BSL 140 mm and an estimated body mass of 10 kg; *Gaetano, Abdala & Govender, 2017*) or *Prozostrodon*. Despite the smaller body size, however, the bone tissues of *Irajatherium* exhibit rapid growth rates, similar to the larger *Tritylodon*.

Among the small-sized prozostrodontians from South America, *Irajatherium* is the only one in which fossorial features in the humerus have been recognized (*Martinelli et al., 2005*). This taxon has a stout humerus, with twisted and broad proximal and distal epiphyses, a proximally deep bicipital groove, and two well-developed processes for the teres major muscle (*Martinelli et al., 2005*; *Oliveira, Martinelli & Soares, 2011*). The osteohistology revealed a relatively thick cortex where the compact bone wall (cortical thickness) comprises 29% of the bone diameter. Similarly thick cortices have been found in extant burrowing mammals such as *Bathyergus suillus* (*Montoya-Sanhueza & Chinsamy, 2017*) *Heterocephalus glaber* (naked mole rat, 31%) and *Erethizon* (New World porcupine >30%), as well as the burrowing or digging lizards *Gerrhonotus grantis*, *Heloderma suspectum* and *Phrynosoma douglassi* (*Magwene, 1993*; *Botha & Chinsamy, 2004*). A thick, robust humerus would have aided in counteracting strong bending forces during forelimb digging. *Prozostrodon* also has relatively thick cortices (humerus 30%, femur 35%), but no fossorial adaptations can be identified in the limb bones of this taxon. The thick cortices may be attributed to a larger body size compared to *Irajatherium* or as yet unidentified biomechanical adaptations.

The bone tissues of *Brasilodon* and *Brasilitherium* are similar, with a mixture of woven and parallel-fibered bone tissues. If we place all of our brasilodontid study material in an ontogenetic series: the *Brasilitherium* UFRGS-PV-1308-T individual is the smallest, with *Brasilodon* UFRGS-PV-765-T being larger and *Brasilitherium* UFRGS-PV-1043-T the largest individual in the series. This is supported by the osteohistology analysis. The bones of the smallest *Brasilitherium* UFRGS-PV-1308-T contain a mixture of rapidly forming woven-fibered bone and slower growing parallel-fibered bone. The absence of any growth marks or an EFS suggests that this individual was in an early ontogenetic stage (within its first year) at the time of death. The bone tissues of *Brasilodon* UFRGS-PV-765-T also contain a mixture of woven-fibered bone and parallel-fibered bone, but there appears to be an annulus at the bone periphery, suggesting that it was ontogenetically older (it had reached its first year) than the small *Brasilitherium* UFRGS-PV-1308-T. The larger *Brasilitherium* UFRGS-PV-1043-T contains a LAG with continued growth after this growth mark, suggesting that it was ontogenetically older than the *Brasilodon* individual. It is recognized that our sample size is very small and more material is required for osteohistology to be used as supportive evidence for the synonymy of these two taxa.

However, it is worth noting that there is no evidence in the study material to contradict the possibility that *Brasilodon* and *Brasilitherium* are synonymous and that they may represent different ontogenetic stages in a series (see also *Liu & Olsen, 2010*; *Martinelli & Bonaparte, 2011*; *Martinelli, 2017*).

The presence of a mid-cortical LAG in the largest *Brasilitherium* individual (UFRGS-PV-1043-T), prior to the onset of the slower forming peripheral parallel-fibered bone, suggests that this taxon took longer than one year to reach somatic or reproductive maturity. Thus, although it was a tiny species and capable of growing quickly (given the presence of rapidly forming woven-fibered bone), it still took longer than one year to reach asymptotic size. This contrasts to what is seen in similar-sized extant mammals (e.g., rodents; *Nesterenko & Ohdachi, 2001*).

Comparing the brasilodontids with mammaliaforms is difficult because the only non-mammalian mammaliaform that has been sectioned is the tiny (BSL 30 mm, *Kemp, 2005*) *Morganucodon watsoni* (*Chinsamy & Hurum, 2006*) and the sections are poorly preserved. *Chinsamy & Hurum (2006)* identified a patch of woven-fibered bone and noted some peripheral parallel-fibered bone tissue with LAGs in their *Morganucodon* sample, but much of the material is diagenetic. There is another section of a *Morganucodon* ulna in the Muséum national d'Histoire naturelle (Paris, France), but it is also badly preserved and little information can be obtained from the sample (J Botha-Brink, pers. obs., 2015). Given that *Chinsamy & Hurum (2006)* noted a mixture of woven-fibered and parallel-fibred bone, it appears that it grew in a similar manner to the brasilodontids. However, *O'Meara & Asher (2016)* proposed that *Morganucodon* grew more similarly to extant placental mammals than did more basal non-mammaliaform cynodonts because it exhibits a truncated growth period, with rapid juvenile growth and a very short period of adult growth. The narrow range of skull lengths in *Morganucodon* supports their proposal where rapid attainment of somatic maturity would restrict size variation (*O'Meara & Asher, 2016*). The skull length variation of *Brasilodon* and *Brasilitherium* (BSL ranging from 20–40 mm) exceeds that of the known samples of *Morganucodon* (*Luo, Crompton & Sun, 2001*; *O'Meara & Asher, 2016*). Combined with the osteohistology results, it suggests that these Brazilian non-mammaliaform prozostrodontians had a more protracted growth period where, although juvenile growth was rapid, they still took longer to reach skeletal maturity than did *Morganucodon*.

*Brasilodon* and *Brasilitherium* grew more quickly than the similar-sized Late Cretaceous eutherian mammals *Barunlestes butleri* (BSL approximately 30 mm, *Kielan-Jaworowska, Cifelli & Luo, 2004*) and *Zalambdalestes lechei* (BSL approximately 50 mm, *Kielan-Jaworowska, Cifelli & Luo, 2004*). Contrary to the brasilodontids, the bones of these eutherians are composed entirely of very poorly vascularized lamellar-zonal bone, indicative of slow growth (*Chinsamy & Hurum, 2006*). Given that all these taxa were similar in body size, the differences in growth cannot be attributed to differences in size. Instead, the brasilodontids exhibit similar bone tissues and growth patterns to the Late Cretaceous multituberculates *Kryptobaatar dashzevegi* (BSL approximately 30 mm, *Kielan-Jaworowska, 1970*) and *Nemegtbaatar gobiensis* (BSL approximately 45 mm, *Kielan-Jaworowska, 1974*). *Chinsamy & Hurum (2006)* found woven-fibered bone in a *Kryptobaatar* femur from what

appears to be a juvenile individual (absence of decreased growth rate), indicating rapid growth. The femur of *Nemegtbaatar* contains parallel-fibered bone and a LAG, indicating slightly slower cyclical growth. Both bone tissue types and growth marks were observed in the brasilodontids, suggesting a closer similarity in growth pattern to the multituberculates than to the eutherian mammals *Barunlestes* and *Zalambdalestes*. Sampling more early eutherians and other Mesozoic mammal lineages, however, will provide a more conclusive assessment of the growth patterns of stem and early crown mammals.

Although comprehensive studies on extant monotreme and marsupial mammal osteohistology are lacking, brief descriptions of isolated elements show that the extant monotremes *Tachyglossus aculeatus* (echidna) and *Ornithorhynchus anatinus* (platypus), as well as the marsupials *Didelphis virginiana* (North American opossum) and *Macropus* (kangaroo) (*Chinsamy-Turan & Hurum, 2012*) exhibit LAGs during mid-ontogeny showing that they exhibit a protracted growth period during this stage. However, these taxa are all much larger than the brasilodontids, making direct comparisons difficult and the osteohistology of marsupials of equivalent size (e.g., marsupial mice) have yet to be described. Extant placental mammals of a similar body size to the brasilodontids typically take less than one year (just a few weeks or months depending on the species) to reach reproductive maturity. Despite the brasilodontids being similar in size to small extant placental mammals that typically reach reproductive maturity within one year (*Miller & Zammuto, 1983*; *Promislow & Harvey, 1990*), these animals still took more than one year to reach growth attenuation. The relatively more protracted growth period of the brasilodontids suggests that they had yet to attain the truncated growth period postulated for *Morganucodon* (Fig. 1(6)), which is typical of extant placental mammals of similar body size (*O'Meara & Asher, 2016*).

## CONCLUSIONS

The osteohistological assessment of several non-mammaliaform prozostrodontian cynodont taxa from basal to more derived forms indicates the presence of high growth rates, similar to other non-mammaliaform cynodonts. Rapid early growth with annual interruptions later in ontogeny is clearly observed in *Prozostrodon*, the most basal prozostrodontian, and in the tritheledontid *Irajatherium*. *Brasilodon* and *Brasilitherium*, which are currently considered to be the sister taxa to Mammaliaformes, grew more slowly than the less derived cynodonts and more similarly to the mammaliaform *Morganucodon* and the multituberculates *Kryptobaatar* and *Nemegtbaatar*. The slower growth rates in these brasilodontids compared to other non-mammaliaform cynodonts may be related to phylogeny and/or decreased body sizes as all these taxa were very small. When compared with similar-sized extant mammals, they may have grown more slowly to adult size as their osteohistology shows it took more than one year for growth to attenuate, thus taking more than one year for these animals to reach skeletal maturity. Thus, although the Prozostrodontia exhibit increasingly mammalian characteristics, including rapid juvenile growth, the small derived, brasilodontid non-mammaliaform prozostrodontians still exhibit an extended growth period compared to similar-sized extant mammals.

**Institutional abbreviations**

| | |
|---|---|
| **NMQR** | National Museum, Bloemfontein. |
| **UFRGSPVT** | Universidade Federal do Rio Grande do Sul, Brazil, Paleovertebrates Collection, Triassic. |

### Funding

This work was supported by the National Research Foundation (UID 98819, 104688), the Palaeontological Scientific Trust (PAST), Johannesburg, South Africa, and DST-NRF Centre of Excellence in Palaeosciences (CoE-Pal). The funders had no role in study design, data collection and analysis, decision to publish, or preparation of the manuscript.

### Grant Disclosures

The following grant information was disclosed by the authors:
National Research Foundation: UID 98819, 104688.
Palaeontological Scientific Trust (PAST), Johannesburg, South Africa.
DST-NRF Centre of Excellence in Palaeosciences (CoE-Pal).

### Competing Interests

The authors declare there are no competing interests.

### Author Contributions

- Jennifer Botha-Brink conceived and designed the experiments, performed the experiments, analyzed the data, contributed reagents/materials/analysis tools, prepared figures and/or tables, authored or reviewed drafts of the paper, approved the final draft.
- Marina Bento Soares conceived and designed the experiments, contributed reagents/materials/analysis tools, authored or reviewed drafts of the paper, approved the final draft.
- Agustín G. Martinelli contributed reagents/materials/analysis tools, authored or reviewed drafts of the paper, approved the final draft.

### Data Availability

The raw data are included in the manuscript as osteohistology photographs in the figures.

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
