# Peer review of "Osteohistology of Late Triassic prozostrodontian cynodonts from Brazil"

_PeerJ, doi:10.7717/peerj.5029_

## Round 0.1 · original submission · Minor Revisions

Your manuscript “Osteohistology of Late Triassic prozostrodontian cynodonts from Brazil” has now been seen by three reviewers. All of the reviewers agree that the core osteohistological data that you present are unique and important, but a number of issues were raised that will need to be addressed before your study can be considered for publication.

My impression is that the accessibility of the paper could be significantly improved by expanding the introduction to include more background information on the use of osteohistological data in cynodont paleobiology. What has been already been done in this area of study? How were the results interpreted? What are the different bone types that have been identified in various cynodont taxa and what do they tell us? As the manuscript stands, the introduction is very short and the naïve reader will find it difficult to follow the results. Reviewer 3 expressed similar concerns. The sentence starting on line 100, “Life history data…” could be a new paragraph that goes into much more depth on the background of the problem. Among other things, I noticed that “LAG” is never defined.

Though certainly not essential, an image showing what these animals looked like (skulls, skeletons?) would be useful. Figure 5 also gives important information about the relationships and adaptations of the study taxa (6 out of 7 of which do not relate to osteohistology) that would be useful background information as part of the expanded introduction that I have recommended; you might consider making this the first figure of the paper rather than the last figure. Again, this would provide readers with more context before diving into the results, thereby making the paper more accessible to non-experts.

I agree with Reviewer 2 that additional labels would improve Figures 1-4. I do not see the “PFB” label mentioned in the caption of Figure 3 on the figure itself. I also do not see the “C” label for that panel.

Reviewer 3 had significant concerns about the “Implications for non-mammaliaform cynodont physiology” section, arguing that it is “a rather confusing account of the relationship of bone histology to endothermy”. I also found it difficult to connect the dots in some places. I would urge you to consider whether this section could be shortened, perhaps to focus primarily on the inferences that can be gleaned from the osteohistological data.

Reviewers 1 and 2 both provide recommendations for stylistic changes, and I have provided other edits in the attached annotated pdf.

I look forward to see the revised manuscript in due course.

Reviewer 1 ·

Basic reporting

The fossils studied are really important to understand life history and physiology around mammal origins. It is original and important what the authors have done.
This paper treats Brasilodon and Brasilotherium as separate taxa – the standard way – although it hints from unpublished data from some of the authors that they may represent different ontogenetic stages of the same species. The histological data presented supports the latter hypothesis.

The discussion of several taxa in lines 375-382 is in part confusing. I think the ‘However’ in line 378 misleads, as the finding in Irajatherium does no contradict what is reported just before – it just leads to interpretations of some kind, etc. Please revise this section.

In terms of bone compactness and small, burrowing mammals, the authors may wish to consult this work on bone compactness in talpids:
Meier PS, et al (2013) Evolution of bone compactness in extant and extinct moles (Talpidae): exploring humeral microstructure in small fossorial mammals. BMC Evolutionary Biology 13:55

Figure 5 is very useful. It would be good to add in the caption the source of some of those events mapped other than those treated in the manuscript (for example, reference for point 5 on vibrissae). This reference attempted something similar but at a much more general and rigorous level – it may be outdated but the authors may wish to check if there is information there to be integrated or commented upon.
Sánchez-Villagra MR (2010) Developmental palaeontology in synapsids: the rock record of ontogeny in mammals and their closest relatives. Proc Roy Soc London, B. 277: 1139-1147.

The authors may consider adding a table summarizing the histological features described for the different taxa, also those cited from the literature. I do not see this as essential.

In several places of the manuscript, the word ‘interestingly’ is used (e.g., lines 382, 397, 446) – this is a stylistic aspect criticized in many writing manuals with which I concur – I invite the authors to consider this. Here I quote from Letzman 1995: ‘Rather than telling a reader that a result is significant or interesting, show them how it is significant or interesting’. The authors do already the second, so they can just delete the ‘Interestingly’. (Check if you wish a dialogue in ‘Captain fantastic’ which refers to ‘interesting’, www.youtube.com/watch?v=YXimxT2iE8I)

Experimental design

no comment

Validity of the findings

no comment

Additional comments

Line 53 – ‘in its definition’ – it is unclear what is meant here. This sentence presents terms most readers won’t be familiar with – I think you mean that crown mammalia is part of mammaliaformes – correct? Maybe the second sentence of the Introduction, to start, adding the stratigraphic occurrence, is more clear?
Line 56: I would delete ‘unranked’
Line 70 – being this a new paragraph, I would not start with ‘However’. See also line 600.
line 165: …were so small THAT only one or two…
lines 294-295: something missing and comma misplaced.
Line 324: I would rephrase to:..and provides an example of the predominant type of growth… - is this what you mean?
Line 325: …well vascularized, COMMA
Line 341: …all of which express…
Line 347 …continued, COMMA
Line 352 …growing, COMMA
Line 703-794 – format italics journal.

There is a paper by Felder et al. 2018 that just came out on animal’s size and secondary osteons which may be relevant to consider in the discussion in this work on scaling.
Felder AA, Phillips C, Cornish H, Cooke M, Hutchinson JR, Doube M. 2017 Secondary osteons scale allometrically in mammalian humerus and femur. R. Soc. Open sci. 4: 170431. http://dx.doi.org/10.1098/rsos.170431

·

Basic reporting

The manuscript is overall well written; the language is clear and unambiguous. The authors take great care in explaining terms which is very helpful for following the results and discussion. I only noticed a few typos in line 165 “than” should be replaced with “that”; line 242 “there” should be replaced with “it”, line 607 “Morganucodon (Fig. 5(6)), that which is typical of extant placental” delete that or which.

The figures are relevant and of high quality. The raw data in form of images is already included in the manuscript. However, I would suggest the following minor edits

1. Figures 1–4: to better relate the structures described in the text with the images it would be helpful to label examples of e.g., Fig. 1 large resorption cavities, vascular canals, primary osteons; Fig. 2 primary osteons, radiating canals osteocyte lacunae and so on.

2. Figure 5: Please explain the color pattern (change from yellow to red, why are some taxa white and others black) in the figure caption. I would further suggest replacing clade names with the names of the species mention in the discussion (for example Procynosuchus, Triarachodon, Massetognathus). Though the figure might take up a bit more space it would make it easier to relate the section on implications for cyndodont physiology to the figure

3. Figure 2: LZB is not listed in the abbreviations

Experimental design

no comment. The manuscript meets the standards listed: the research question is well defined, the authors specifically point to the gap in knowledge that this research is filling, the bone histology is described in great detail, the methods are sufficiently detailed.

Validity of the findings

no comment. The manuscript meets the standards

Additional comments

1. The authors are in few instances vague in their use of derived prozostrodontians and mammaliaform. For example:

line 58 “the Tritylodontidae and Tritheledontidae, both closely related to the Mammalia”
it would be more correct to use Mammaliaformes here as the sister clade.

line 427 “because the only mammaliaform that has been sectioned is the...” as all extant mammals are mammaliaforms this statement is not correct. I do see what the authors mean; but it would be more appropriate to use non-mammalian mammaliaform or basal mammaliaform

line 48, 630 “small derived prozostrodontians” I suggest changing this to non-mammaliaform prozostrodontians. As derived prozostrodontians could refer to any derived mammal today as well.

There might be other instances and I believe it would be good to check the manuscript carefully. It is all too easy to slip into using mammaliaforms or prozostrodontians as if it would only refer to the basal forms within this group.

2. Growths and life span of Morganucodon. New, but unpublished data (presented for example at the International Symposium on Paleohistology 2015 by Newham et al.) on cementum suggest that Morganucodon might have grown for much longer time periods. Even Chinsamy and Hurum (2006) indicated that several LAG’s were visible in the ulna of Morganucodon though they do not mention or indicate how many. I see that this will be fairly difficult to address as the cementum data is not published yet, but it might be possible to include Chinsamy and Hurum’s (2006:p.328) statement on Morganucodon: “This area includes several rest lines, which indicate pauses in the rate of bone formation, and hence, pauses in growth.” Though the authors note that brasilodontids had a relatively more protracted growth period, it seems possible that it might have been even longer for Morganucodon.

3. line 43 Late Triassic/Early Jurassic mammaliaform Morganucodon and Late Cretaceous multituberculate mammals. I would suggest to change “and Late Cretaceous multituberculate mammals” to “and the Late Cretaceous multituberculates Nemegtbaatar and Kryptobaatar” to be more specific and consistent with the reference to Morganucodon

Reviewer 3 ·

Basic reporting

See below

Experimental design

See below

Validity of the findings

See below

Additional comments

Botha Brink and her co-workers are the leaders when it comes to bone histology of therapsids.

This paper gives a detailed account of the histology of Late Triassic forms that are mammals closest relatives. The account given of the histology is a valuable contribution and I recommend publication of this portion of the paper. However the authors expand far beyond the important descriptive part and provide a rather confusing account of the relationship of bone histology to endothermy.

I think it would be improved if they clarify the ontogeny of different bone types in these advanced brazilosdontids and compare this to mammals of the same size.

As understand their conclusion the first type of bone formed is FLB indicating rapid growth this is followed by PFB that includes a lag line and this in turn by FLB and always lack a EFS and lack definitive growth? They interpret the single lag to mean that after one year only a slight amount growth occurs. It would be helpful if more attention was paid to the meaning of these layers. The authors suggest that FLB = rapid growth, PFB = slow growth. Is the latter due to seasonal changes? And if so, does the lag line occur sometime during this season when adverse environmental conditions prevent growth completely and there may be some bone resorption? Following this does some slow growth occur as conditions improve? It would be helpful if this could be compared with the same detail in a small mammal. Lag lines are common in marsupials, but do the small mammals they are comparing their specimens with show a single lag line followed by FLB and EFS. They conclude that many of the attributes of homoeothermic endothermic was present in the forms the authors studied and claim they possibly possesed elevated maximum and basal metabolic rates. I suggest a few of these should be included . Definitive growth occurs in small mammals; this is associated with endothermy but I appreciate the authors caution in not claiming this for prozostrodonts.

The value of this paper is the detailed description of the bone histology of prozostrodontis and the conclusion on their growth patterns. I think the long sections on the origin of endothermy detract rather than add to the papers importance. For example the relative size of the olfactory chamber in Thrinaxodon is similar to all the non-mamalian cynodonts, I agree that some reptiles use limited endogenous heat production during parental, but there is no credible basis for the claim that Lumkuia or other advanced forms, before mammals had extended parental care. Also there is no evidence that Morganucodon possessed ossified turnbinals. Some of these may be true but without some justification they add little to what I consider to be an excellent paper.

---

## Round 0.2 · accepted · Accept

Thank you for your close attention to the reviewers' comments and suggestions, and for the substantial revisions that you have made to the original text. Your paper has been significantly improved by these changes, and I now consider it to be suitable for publication in PeerJ without any additional revision. Thanks, again, for this very interesting submission, and I look forward to seeing it in print!

#